# miR-214 aggravates oxidative stress in thalassemic erythroid cells by targeting *ATF4*

Tipparat Penglong[1], Apisara Saensuwanna[2], Husanai Jantapaso[2], Pongpon Phuwakanjana[3,4], Natee Jearawiriyapaisarn[3], Kittiphong Paiboonsukwong[3], Worrawit Wanichsuwan[5], Kanitta Srinoun[2]*

1 Department of Pathology, Faculty of Medicine, Prince of Songkla University, Hat Yai, Songkhla, Thailand, 2 Faculty of Medical Technology, Prince of Songkla University, Hat Yai, Songkhla, Thailand, 3 Thalassemia Research Center, Institute of Molecular Biosciences, Mahidol University, Nakhon Pathom, Thailand, 4 Department of Biochemistry, Faculty of Medicine Siriraj Hospital, Mahidol University, Bangkok, Thailand, 5 Medical Science Research and Innovation Institute, Research and Development Office, Prince of Songkla University, Hat Yai, Songkhla, Thailand

* Kanitta.s@psu.ac.th

**Data Availability Statement:** All relevant data are within the paper and its Supporting information files.

## Abstract

Oxidative damage to erythroid cells plays a key role in the pathogenesis of thalassemia. The oxidative stress in thalassemia is potentiated by heme, nonheme iron, and free iron produced by the Fenton reaction, due to degradation of the unstable hemoglobin and iron overload. In addition, the levels of antioxidant enzymes and molecules are significantly decreased in erythrocytes in α- and β-thalassemia. The control of oxidative stress in red blood cells (RBCs) is known to be mediated by microRNAs (miRNAs). In erythroid cells, microR-214 (miR-214) has been reported to respond to external oxidative stress. However, the molecular mechanisms underlying this phenomenon remain unclear, especially during thalassemic erythropoiesis. In the present study, to further understand how miR-214 aggravates oxidative stress in thalassemia erythroid cells, we investigated the molecular mechanism of miR-214 and its regulation of the oxidative status in thalassemia erythrocytes. We have reported a biphasic expression of miR-214 in β- and α-thalassemia. In the present study the effect of miR-214 expression was investigated by using miR -inhibitor and -mimic transfection in erythroid cell lines induced by hemin. Our study showed a biphasic expression of miR-214 in β- and α-thalassemia. Subsequently, we examined the effect of miR-214 on erythroid differentiation in thalassemia. Our study reveals the loss-of-function of miR-214 during translational activation of activating transcription factor 4 mRNA, leading to decreased reactive oxygen species levels and increased glutathione levels in thalassemia erythroid cell. Our results suggest that the expression of activating transcription factor 4 regulated by miR-214 is important for oxidative stress modulation in thalassemic erythroid cells. Our findings can help to better understand the molecular mechanism of miRNA and transcription factors in regulation of oxidative status in erythroid cells, particularly in thalassemia, and could be useful for managing and relieving severe anemia symptoms in patients in the future.

**Funding:** This work was supported by the National Science, Research and Innovation Fund (NSRF), Prince of Songkla University (Grant No. MET6505130S), the Faculty of Medical Technology, Prince of Songkla University, and Mahidol University (Basic Research Fund: fiscal year 2021)]. The funders had no role in study design, data collection and analysis, decision to publish, or preparation of the manuscript.

**Competing interests:** The authors have declared that no competing interests exist.

## Introduction

α- and β-thalassemia are a heterogenous group of hereditary blood disorders caused by defects in α- and β-globin chain production due to mutations in the globin genes. Impairment of globin chain synthesis results in an accumulation of $\beta_4$- or aggregation of highly oxidative unpaired α-globin chains that comprise hemichromes, free iron, and nonheme iron. These compounds aggregate on red blood cell membranes and produce reactive oxygen species (ROS) through the Fenton reaction [1–4]. The oxidative environment produced by these free radicals induces ineffective erythropoiesis and red blood cell hemolysis, which eventually promotes severe anemia and other pathophysiologies [5, 6].

MicroRNAs (miRNAs) are endogenous short noncoding RNAs (20–23 nucleotides) that regulate gene expression by targeting specific mRNAs for cleavage or translational repression by binding to specific sequence in the 3' untranslated region of target mRNA target [7, 8]. Previous studies have revealed that miRNAs play important roles in regulating oxidative stress in both, healthy and diseased individuals [9, 10]. Sangokoya *et al.* [11] demonstrated that miR-144 directly inhibits the mRNA target, nuclear factor-erythroid 2-related factor 2 (NRF2), a well-known transcription factor involved in modulating antioxidant production in sickle cell disease [11]. NRF2 suppression is associated with decreased glutathione (GSH) regeneration. Our previous study reported that high miR-144 expression contributes to the suppression of NRF2 expression in thalassemia. miR-144 expression is correlated with GSH levels and the anemia severity, particularly in thalassemia. Reduced miR-144 expression promotes high expression of NRF2 and GSH, whereas overexpression of miR-144 leads to a significant increase in $H_2O_2$ [12]. We also explored the expression of miR-214 in the reticulocytes of α- and β-thalassemia samples. We found that the expression of miR-214 increased in HbH disease; however, it was not significantly different from β-thalassemia/HbE reticulocytes. Furthermore, the expression level of miR-214 was positively associated with ROS levels. A reduction in the mRNA of activating transcription factor 4 (ATF4) was observed in both, HbH (β4) disease, and β-thalassemia/Hb E thalassemia groups [13]. ATF4 is a crucial transcription factor that responds to cellular stress. ATF4 expression is mediated by heme-regulated eIF2alpha kinase (HRI)/eIF2α signaling which upregulates its translation in response to oxidative conditions [14–16]. However, the function of miR-214 and its mRNA target in oxidative stress remains unclear, especially during thalassemic erythropoiesis. In this study, we investigated the molecular mechanism of miR-214 and its regulation of the oxidative status in thalassemia erythrocytes in an attempt to gain insight into how miR-214 exacerbates oxidative stress in thalassemia erythroid cells. Our results demonstrate expression of miR-214 in β-and α- thalassemia erythroid cell. We also studied the effect of miR-214 expression by using miR-inhibitor and miR-mimic transfection in erythroid cell lines and erythroid differentiation in thalassemia.

## Materials and methods

### Participants

Peripheral blood samples were collected from healthy donors (50 mL; n = 3) and from patients with HbH disease and β-thalassemia/HbE (25 mL; n = 3). This study was conducted in accordance with the principles of the Declaration of Helsinki, and all procedures were approved by the Ethics Committee of Nakhon Pathom Hospital, Nakhon Pathom, Thailand (Approval NPH-REC No. 001/2021) from October 2022 to January 2023. Written informed consent was obtained from all the participants.

## Human erythroid progenitor cell culture

Briefly, peripheral blood mononuclear cells (PBMCs) were separated by density gradient centrifugation using Lymphoprep™ (Axis-Shield, Oslo, Norway); the density was 1.077 g/mL and CD34+ hematopoietic stem/progenitor cells (HSPCs) were purified using immunomagnetic-positive selection using the CD34 MicroBead Kit (Miltenyi Biotec, Gladbach, Germany) with a cell separation column. The isolated CD34+ HSPCs were cultured in erythroid differentiation medium (EDM) [17]. Briefly, purified CD34+ HSPCs were expanded in X-VIVO 15 (Lonza, Basel, Switzerland) supplemented with 100 ng/mL human stem cell factor (SCF; PeproTech, Rocky Hill, NJ),), 100 ng/mL human thrombopoietin (TPO; PeproTech), 100 ng/mL recombinant human Flt3-ligand (Flt3-L; PeproTech), 1X GlutaMAX™ (GIBCO-Invitrogen, NY, USA), and 100 U/mL penicillin/streptomycin (GIBCO-Invitrogen) for 4 d to a density of $10^5$ cells/mL. CD34+ HSPCs were differentiated toward the erythroid lineage in a three-phase liquid culture system using three different erythroid differentiation culture media. The basal medium consisted of Iscove's Modified Dulbecco Medium (IMDM; HyClone, Logan, UT) supplemented with 5% human AB plasma, 10 μg/mL recombinant human insulin (Novo Nordisk A/S, Bagsvaerd, Denmark), 2 U/mL heparin (LEO Pharma A/S, Ballerup, Denmark), 330 μg/mL holo-human transferrin (ProSpec, Rehovot, Israel), 3 U/mL erythropoietin (EPO; Janssen-Cilag, Bangkok, Thailand), 1X Gluta-MAX™, and 100 U/mL penicillin/ streptomycin. During days 0–7 of the culture (EDM I), the basal medium was further supplemented with 100 ng/mL human SCF, 5 ng/mL human interleukin 3 (IL-3; PeproTech), and 1 μM hydrocortisone (Sigma-Aldrich, St. Louis, MO). During days 7–11 of the culture (EDM II), the basal medium was supplemented with 100 ng/mL human SCF only. During days 11–16 of the culture (EDM III), the cells were cultured in the basal medium without additional supplements. During the culture, the cells were maintained at $3–5 \times 10^5$ cells/mL, $5–8 \times 10^5$ cells/mL, and $1–2 \times 10^6$ cells/mL in EDM I, II, and III, respectively. Cells were incubated in 5% $CO_2$ at 37 ˚C.

## miR-214 mimic and anti-miR-214 inhibitor transfection

On day 7 of the culture, erythroid cells ($4 \times 10^5$ cells) were treated with 80 nM miR-214 miRNA mimic and negative mimic control (mirVana™ miRNA mimic; Applied Biosystems, Waltham, MA, USA) to increase the expression of miRNA. The miR-214 inhibitor with 80 nM (Anti-miR™ miRNA Inhibitor; Applied Biosystems) was used to inhibit miRNA expression. Both the miRNA mimic and inhibitor were transfected into erythroblastic cells using Lipofectamine™ RNAiMAX (Invitrogen, Waltham, MA, USA). Cells were collected 24 and 48 h after transfection and expression of miR-214 and mRNA target genes was analyzed using quantitative reverse transcription PCR (qRT-PCR). ROS and GSH levels were evaluated using flow cytometry.

## Cell culture

K562 cells were cultured in Roswell Park Memorial Institute (RPMI) 1640 medium (GIBCO-Invitrogen) with 10% heat-inactivated fetal bovine serum (GIBCO-Invitrogen), 2 mM glutamine, and penicillin/ streptomycin under 5% $CO_2$ and 37 ˚C. To induce oxidative stress, the cells were treated with hemin (Sigma-Aldrich, Missouri, USA). The K562 cells ($3 \times 10^5$ cells) were treated in 24 well plates with 25–75 μM hemin alone or pretreated with 40 nM human mature miR-214 or negative control mimic (mirVana™ miRNA mimic; Applied Biosystem) for 24 h using the Lipofectamine™ RNAiMAX transfection protocol (Invitrogen) followed by 50 μM hemin for 48 h [18]. Cell viability was evaluated using the trypan blue exclusion assay. The cells were harvested and investigated for ROS, GSH, apoptosis, miRNA, and gene expression.

## miRNA and mRNA target gene expression

Total RNA was extracted from cells using the Hybrid-R™ miRNA Isolation Kit (GeneAll, Seoul, South Korea) according to the manufacturer's protocol. miR-214 expression was conducted using TaqMan® Small RNA Assays (Applied Biosystems). miRNA quantitation was performed using a LightCycler® 480 PCR System (Roche Applied Science, Mannheim, Germany). The expression of miRNA normalized against RNU48 expression, and it was calculated using the $2^{-\Delta\Delta Ct}$ (comparative Ct) method. Untreated miR-214 mimic or anti-miR-214 inhibitor as a control. All experiments were performed in triplicates.

For the identification of the *ATF4* gene, expression of the candidate *ATF4* gene targets were determined by RT-qPCR. cDNA was synthesized using one step qRT-PCR (qPCRBIO SyGreen 1-step Lo-ROX, PCR Biosystems Ltd., London, UK) with specific primers, i.e., *ATF4*: forward primer, 5′-ATG ACC GAA ATG AGC TTC CTG-3′ reverse primer, 5′-GCT GGA GAA CCC ATG AGG T-3′ and *GAPDH*: forward primer, 5′-GAA GGT GAA GGT CGG AGT C-3′, reverse primer, 5′-GAA GAT GGT GAT GGG ATT TC-3′. The *GAPDH* as internal control normalizer for relative values were calculated ($2^{-\Delta\Delta Ct}$) and compared among samples. The experiments were carried out in triplicates.

## Flow cytometric analysis of erythroid cell oxidation

To study the oxidative stress status, intracellular ROS and GSH levels were evaluated using flow cytometry. To measure the ROS level, cells were collected and 1 mL of phosphate-buffered saline (PBS) was added, followed by 0.4 mM of 2'-7'-dichlorofluorescin (DCF) diacetate (Sigma-Aldrich) dissolved in 500 μL of methanol. The cells were then incubated at 37 ˚C in a 5% $CO_2$ atmosphere for 15 min. Subsequently, the cells were washed, spun down, and resuspended with 500 μL of PBS before they were transferred to a polystyrene tube. Thereafter, red blood cells were stimulated with 2 mM of freshly prepared $H_2O_2$ and incubated at 25˚C for 20 min [19, 20]. The wavelengths of excitation (Ex) and emission (Em) for dichlorodihydrofluoroscein (DCF) determination by flow cytometry (BD FACSCalibur™, BD Bioscience) were 498 and 522 nm, respectively. The ROS levels in K562 cells were measured using flow cytometry (BD FACSCalibur™, BD Bioscience) and calculated as mean fluorescence using CellQuest software (BD Bioscience). In the human erythroid progenitor cell culture experiment, the ROS levels were measured using BD Accuri™ C6 Plus Flow Cytometry (BD Bioscienc) and analyzed using FlowJo software v10 (FlowJo LLC, BD Bioscience).

GSH levels were determined using the Thiol Green Dye method. Cells stained with Thiol Green Dye can be visualized with a flow cytometer at Ex/Em = 490/520 nm. After collecting, the cells were spun down and resuspended in 1 mL of warm media, followed by addition of 5 μL of 200X Thiol Green Dye (Abcam, Cambridge, United States) in the cell solution, and incubated at 37 ˚C in a 5% $CO_2$ atmosphere for 30 min. Finally, the cells were centrifuged at 1,000 $g$ for 5 min and resuspended in Assay Buffer or PBS. GSH levels in the cell lines were measured using flow cytometry (BD FACSCalibur, BD Bioscience) and determined as mean fluorescence using CellQuest software. GSH levels in human erythroid progenitor cell cultures were determined using a BD Accuri™ C6 Plus Flow Cytometer (BD Bioscience) and analyzed.

## Flow cytometric analyses of erythroid cell apoptosis

K562 cells ($1 \times 10^5$ cells/well) were harvested and stained with Annexin V conjugated to fluorescein isothiocyanate (FITC) and propidium iodide (PI) (BD Biosciences) according to the manufacturer's protocol. The flow cytometry wavelengths used to detect FITC and PI were Ex/Em = 488/530 nm and Ex/Em = 488/>575 nm, respectively. The percentage of apoptotic cells was determined using a BD FACSCalibur (BD Biosciences) and analyzed using the CellQuest

software. In each experiment, unstained cells were used to establish the background fluorescence in the experimental samples.

## Statistical analyses

Data were analyzed using Statistical Package for the Social Sciences, version 23 (SPSS Inc., Chicago, IL, USA). Comparisons between parameters were conducted using the non-parametric Mann–Whitney *U* Test. *P*-values < 0.05 were considered statistically significant.

## Results

### Expression of miR-214 and its target gene, *ATF4*, after treatment with different concentrations of hemin

To determine miR-214 and ATF4 expression under oxidative stress, cells were treated with different concentrations of hemin for 48 h, and cellular miRNA and mRNA expression levels were determined using qRT-PCR. We found a significant dose-dependent four-, six-, and eight-fold increase of miR-214 expression at 25, 50, and 75 μM hemin concentrations, respectively (Fig 1A). We also identified a putative target gene, *ATF4*. We found that the expression of *ATF4* was significantly increased by 10-, 21-, and 17-fold with modulation of hemin concentrations (Fig 1B). Interestingly, *ATF4* expression was highest at the 50 μM hemin concentration.

### Cell oxidative stress after treatment with different concentrations of hemin

To find the most suitable hemin concentration for mimicking the oxidative stress condition in thalassemic erythroid cells and examine the effect of hemin on oxidative stress status, K562 cells were treated with 25, 50, and 75 μM concentrations of hemin. Cell viability was examined using trypan blue staining after 48 h of hemin treatment. The results revealed that the trypan blue-negative cell number was significantly decreased with higher hemin concentrations when compared to untreated cells (Fig 2A). Moreover, DCF and GSH thiol green staining, indicating ROS and GSH levels, respectively, were measured using flow cytometry. The results showed that ROS levels were significantly upregulated by four-, five-, and six-fold with higher hemin

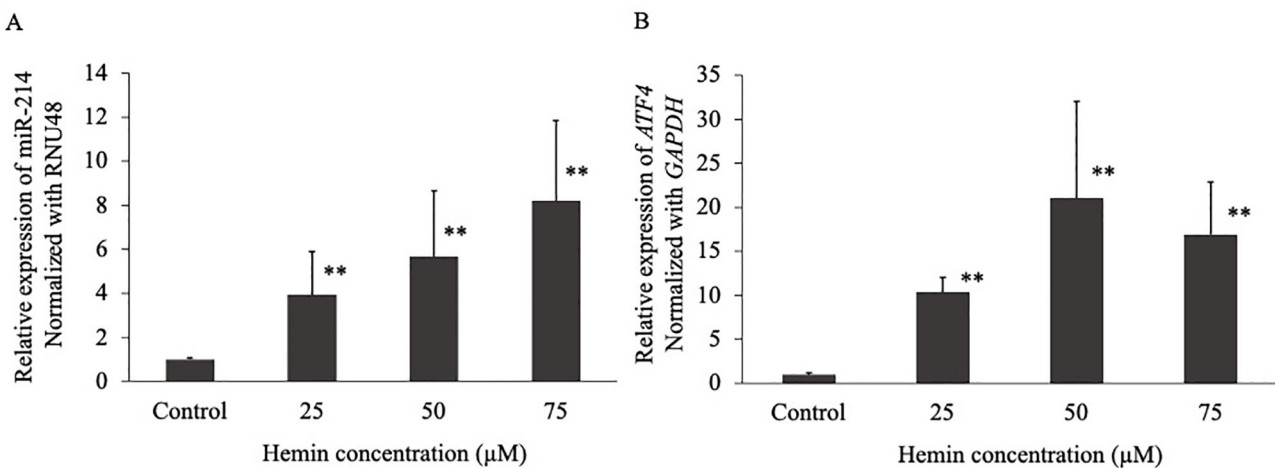

**Fig 1. The expression of miR-214 (A) and the activating transcription factor 4 (*ATF4*) target gene (B) after treating K562 cells with 25, 50, and 75 μM of hemin for 48 h.** Expression was measured using quantitative reverse transcription polymerase chain reaction. **$p < 0.01$, *$p < 0.05$.

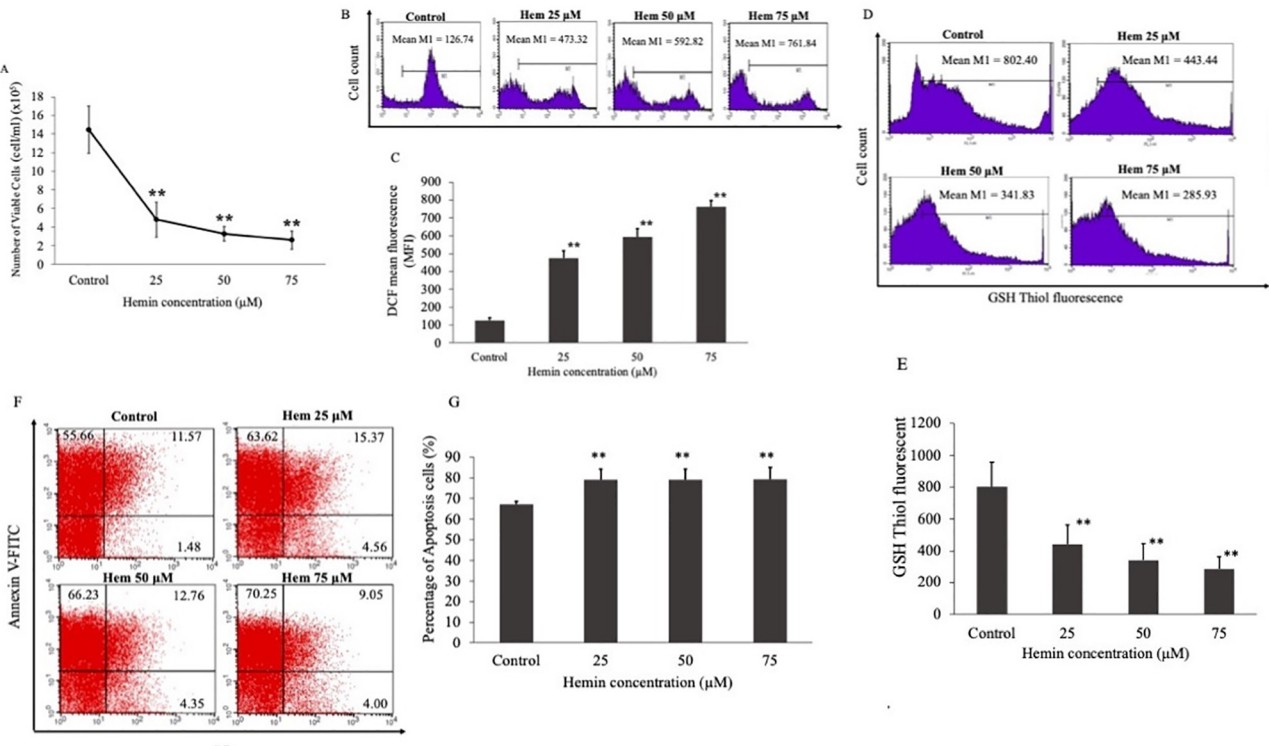

**Fig 2. K562 cells treated with 25, 50, and 75 μM of hemin for 48 h.** (A) Cell viability of K562 cells after hemin treatment. 2'-7'-dichlorofluorescin (DCF) mean florescence of K562 cells after hemin treatment determined by flow cytometry. The data are shown using a (B) mean fluorescent intensity histogram and (C) bar chart. Glutathione (GSH) thiol green florescence of cells after hemin treatment determined using flow cytometry. Data are shown as a (D) mean fluorescent intensity histogram and (E) bar chart. Annexin V and propidium iodide staining of K562 cells after hemin treatment detected using flow cytometry shown in a (F) dot plot diagram and (G) bar chart. **$p < 0.01$, *$p < 0.05$.

concentrations when compared to untreated cells (Fig 2B and 2C). Conversely, GSH levels were significantly downregulated by two-, two-, and three-fold at 25, 50, and 75 μM hemin concentrations, respectively (Fig 2D and 2E). The effect of hemin on apoptosis was assessed using Annexin V and PI staining. Moreover, the apoptosis rate dose-dependently rose from 55.66% (untreated) to 63.62, 66.23, and 70.25 at 25, 50, and 75 μM hemin concentrations, respectively (Fig 2F and 2G).

## Upregulated miR-214 expression in the erythroid cell increased response to oxidative stress

To examine the mechanism of miR-214 in the regulation of oxidative stress in erythroid cells, the expression levels of miR-214 in K562 cells transfected with miR-214 mimics were measured, with or without hemin treatment. The level of miR-214, after transfection with miR-214 mimics alone, dramatically increased by more than 1660000-fold compared to untreated cells. The miR-214 expression level in cells transfected with miR-214 mimics followed by hemin treatment showed a significant 2.8-fold increase compared with cells transfected with miR-214 mimics alone. Moreover, under oxidative stress, the expression level of miR-214 in cells transfected with miR-214 mimics was significantly higher than that in cells transfected with negative mimic controls by approximately 167729-fold (Fig 3A). The expression of *ATF4* was investigated after transfection with miR-214 mimics, with or without hemin treatment. The

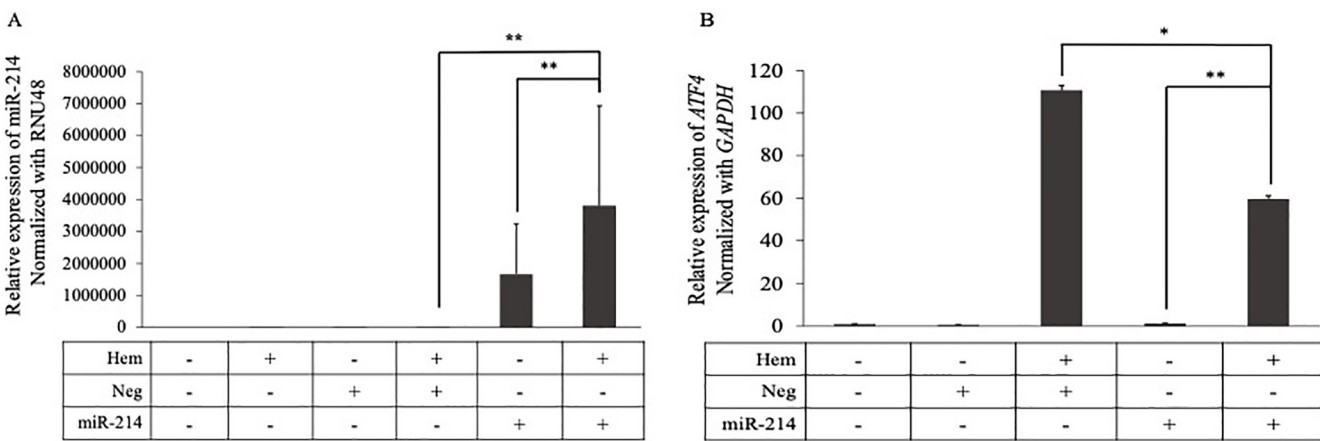

**Fig 3. Relative expression of (A) miR-214 and (B) the *ATF4* target gene after transfection with 40 nM of miR-214 mimics or a negative mimics control for 24 h, followed by treatment with 50 μM of hemin for 48 h in K562 cells.** The expression was measured by quantitative reverse transcription polymerase chain reaction. $^{**}p < 0.01$, $^{*}p < 0.05$.

results showed that the expression of *ATF4* in cells treated with miR-214 and hemin was significantly upregulated (60-fold) compared to that in cells treated only with miR-214, confirming that hemin could induce *ATF4* transcription. Interestingly, under oxidative stress induced by hemin, *ATF4* expression in miR-214 transfected cells were significantly downregulated by two-fold compared to that in the negative mimic control-treated cells (Fig 3B).

We further confirmed the miR-214 overexpression effect on the oxidative stress status. Using flow cytometry, ROS and GSH levels were measured. The DCF mean fluorescence of miR-214 mimic-transfected cells under hemin-induced oxidative stress conditions was significantly higher (3.6-fold) than that of cells transfected with only miR-214 mimics, confirming that hemin could induce ROS production. However, cells treated with only miR-214 did not show altered mean fluorescence when compared to untreated cells. Under hemin conditions, the DCF mean fluorescence in the cells transfected with miR-214 was elevated compared to that in the negative control, though not significantly ($p = 0.14$) (Fig 4A and 4B). For the GSH level, the miR-214-hemin treated cells displayed a significant two-fold decrease in thiol green fluorescence compared to the miR-214 treated cells alone, confirming that hemin could reduce GSH production. However, the level of GSH did not change significantly between cells that were treated with miR-214 mimics and the negative mimic control under oxidative conditions, and cells treated with only miR-214 mimics did not show a significantly different GSH level when compared with untreated cells or negative mimic control-transfected cells alone (Fig 4C and 4D).

To study miR-214 function in apoptosis under oxidative stress conditions, after transfecting cells with miR-214 mimics or negative control mimics followed by hemin treatment, the apoptotic cells were investigated using flow cytometry after staining with Annexin V and PI. Our results demonstrated that the miR-214 mimics with hemin treated cells showed higher apoptosis levels than cells treated with miR-214 mimics alone ($p = 0.091$). However, there was no significant difference in the apoptosis cells between miR-214 and the negative mimic control in cells with hemin-induced oxidative stress (Fig 4E and 4F).

Our data demonstrated that under oxidative conditions, the levels of miR-214 increased; however, oxidative stress could also activate *ATF4*, which is a putative mRNA target gene of miR-214. Further studies on oxidative stress regulation by miR-214 were performed by

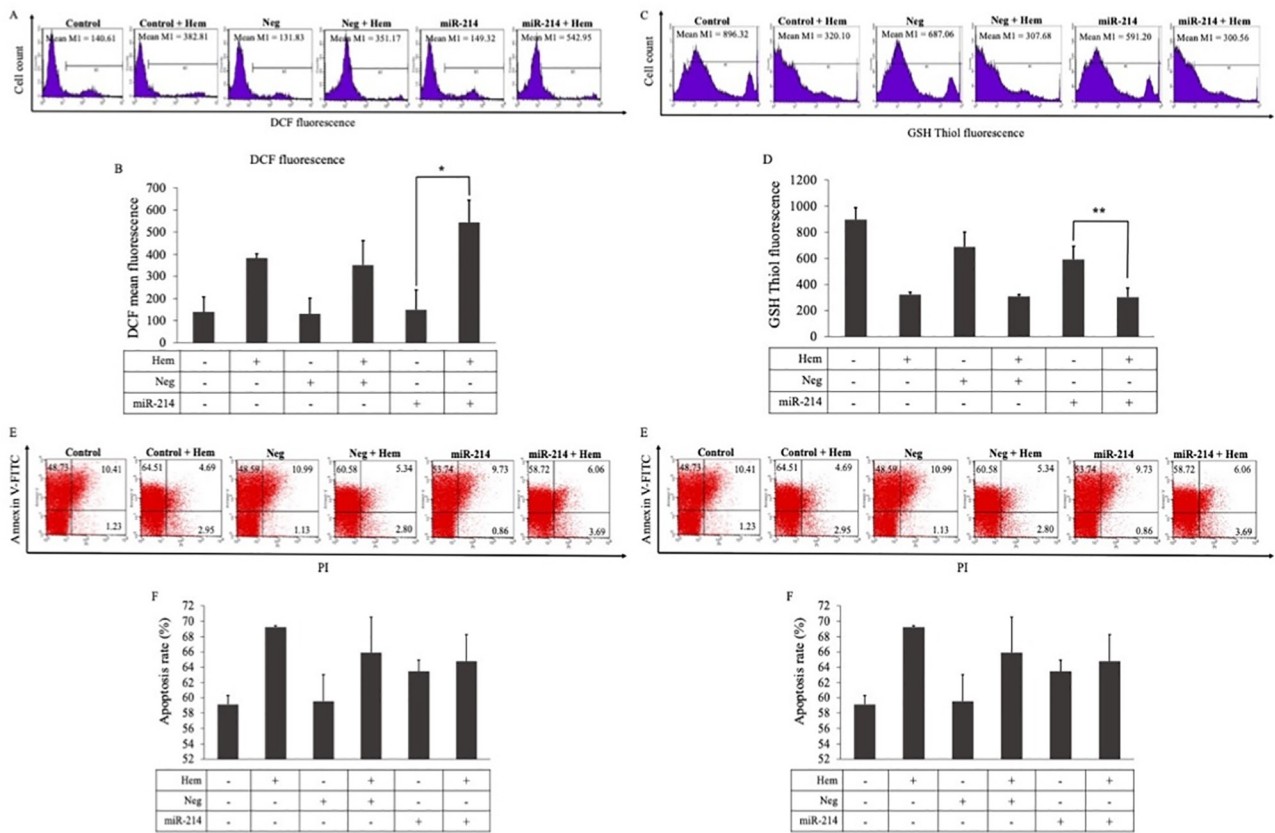

**Fig 4. K562 cells were treated with 40 nM of miR-214 mimics or a negative mimics control for 24 h, followed by treatment with 50 μM of hemin for 48 h.** DCF mean florescence of K562 cells was determined using flow cytometry. Data are shown using a (A) mean fluorescent intensity histogram and (B) bar chart. The glutathione (GSH) thiol green florescence of cells was determined using flow cytometry. The data are shown in a (C) mean fluorescent intensity histogram and (D) bar chart. The Annexin V and propidium iodide staining of K562 cells was detected using flow cytometry. The data are shown in a (E) dot plot diagram and (F) bar chart. **p < 0.01, *p < 0.05.

transfecting miR-214 into cells with hemin-induced oxidative stress, and we revealed that miR-214 could inhibit ATF4 expression under oxidative stress.

## The expression of miR-214 during erythroid differentiation

To investigate the role of miR-214 in oxidative stress in erythroid cells, we examined miR-214 expression in cultured thalassemia erythroid cells and compared it with that in normal cells (S1 Fig). In healthy subjects, the expression of miR-214 was up-regulated during erythroid differentiation (Fig 5). Remarkably, biphasic expression with transient upregulation of miR-214 was detected on day 5 during erythroid differentiation in HbH and β-thalassemia/HbE. The relative expression levels of miR-214 on day 5 in patients with HbH disease (4.5) and β-thalassemia/HbE (4.6) were higher than those in healthy subjects. The level of miR-214 expression in patients with HbH disease and β-thalassemia/HbE was lowest on day 9 and it then increased in a display similar to that of normal subjects during erythroid differentiation. However, in patients with thalassemia, the expression of miR-214 was significantly higher than that in healthy subjects during erythroid differentiation, except on day 9 (Fig 5). On day 7, the relative expression levels of miR-214 in HbH disease were higher than those of the β-thalassemia/HbE group ($p < 0.05$).

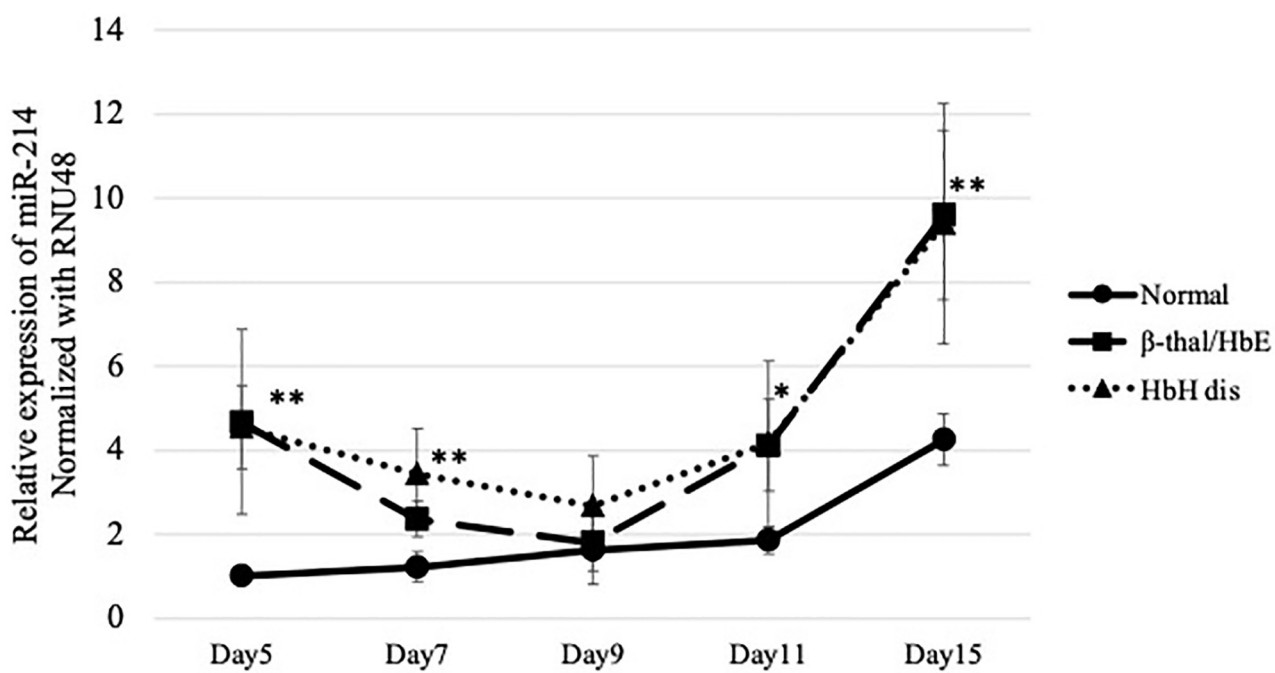

**Fig 5. Expression of miR-214 was normalized by the expression of RNU48 during erythropoiesis in normal subjects and patients with HbH disease and β-thalassemia/HbE.** The experiments were carried out in triplicates. **$p < 0.01$, *$p < 0.05$.

### miR-214 expression aggravates oxidative stress

We also investigated whether upregulation of miR-214 contributes to the induction of oxidative stress in normal tissues. The levels of miR-214 expression dramatically increased after transfection with the miR-214 mimic compared to those in the untreated and negative control groups (Fig 6A) (S1 Table). Loss-of-function was evaluated using an anti-miR-214 inhibitor in β-thalassemia/HbE and HbH diseases during erythroid differentiation. The results demonstrated that transfection of erythroid cells with an anti-miR-214 inhibitor caused a significant reduction in miR-214 in β-thalassemia/HbE and HbH compared to the untreated and negative control inhibitor (Figs 6B and 6C) (S1 Table). In normal samples, the upregulation of miR-214 significantly reduced ATF4 mRNA expression (Fig 7A), whereas a significant increase in ATF4 mRNA expression was observed in erythroid cells of β-thalassemia/HbE and HbH disease samples, which were transfected with anti-miR-214, compared to that in untreated and negative inhibitor control-treated cells (Fig 7B and 7C).

Additionally, the effect of miR-214 upregulation and downregulation on oxidative stress status was also studied. The upregulation of miR-214 had a significant effect on ROS levels and significantly decreased GSH levels (Fig 8). Conversely, miR-214 downregulation caused significant decrease on ROS levels; furthermore, GSH levels was significantly increased in thalassemic erythroblasts transfected with anti-miR-214 compared to the untreated and negative inhibitor control cells (Fig 9). Taken together, these results indicated that *ATF4* is a valid target gene of miR-214 that regulates oxidative stress.

### Discussion

Oxidative stress in erythroid cells occur in various hematological diseases, particularly thalassemia [21–24]. This harmful process can result in cellular damage and apoptosis, leading to

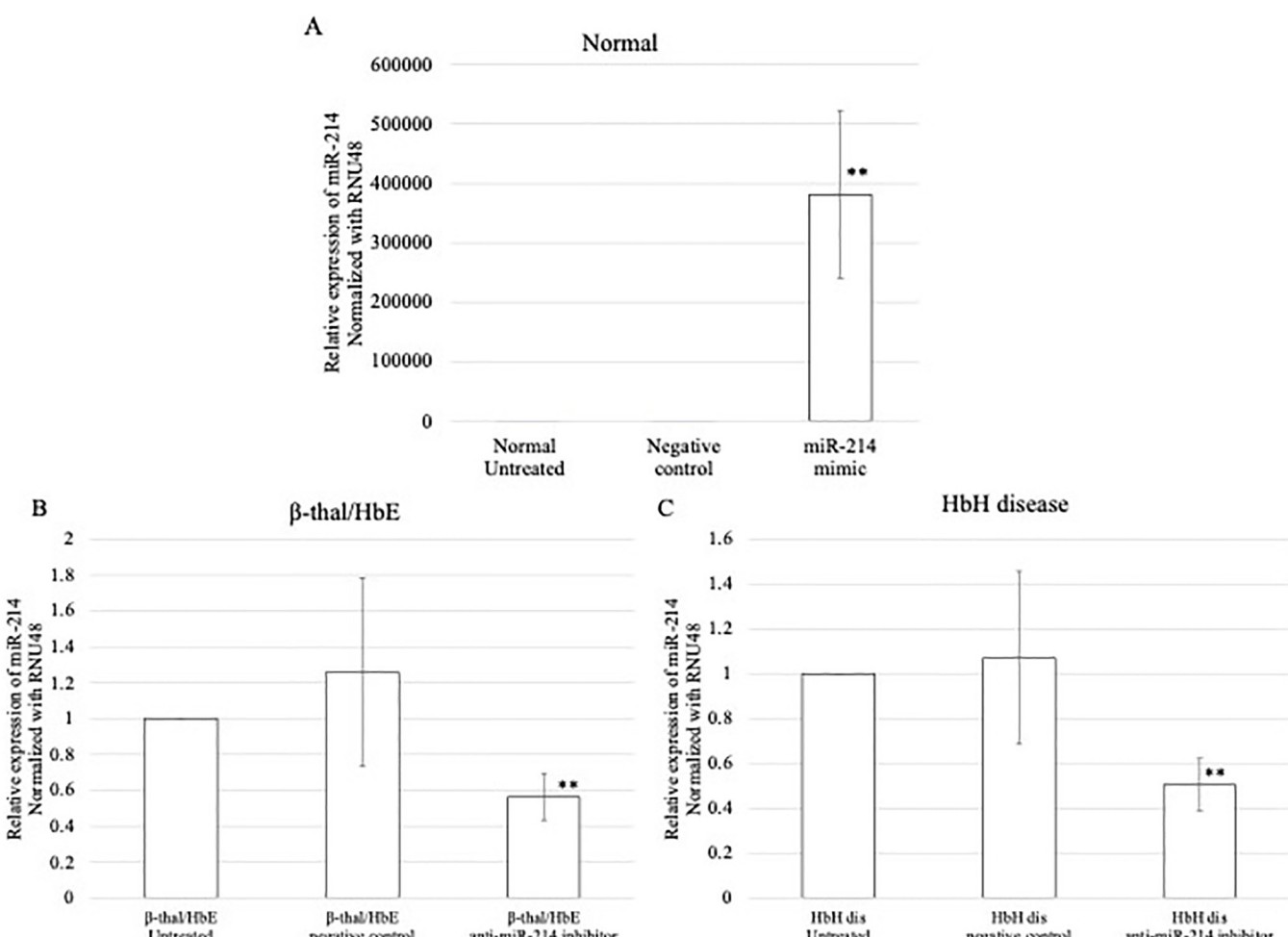

**Fig 6. Quantitative PCR (qPCR) analysis of miR-214 expression after transfection with miR-214 mimic in (A) normal samples and qPCR analysis of the miR-214 expression after transfection with anti-miR -214 inhibitor in (B) β-thalassemia/HbE and (C) HbH disease.** The experiments were carried out in triplicates. \*\*$p < 0.01$, \*$p < 0.05$.

obvious clinical manifestations, such as anemia. Several studies have shown that miRNAs are involved in regulating oxidative stress in many cell types, including RBCs. In this study, we focused on miR-214, which has previously been shown to play a role in oxidative stress in erythroid cells [13, 25]. Our previous study revealed upregulation of miR-214 and downregulation of *ATF4* mRNA target in the reticulocytes of patients with HbH disease. Moreover, we found an association between miR-214 expression, oxidative stress condition, and the severity of anemia [13]. Here, we revealed that the expression of miR-214 is increased in a dose-dependent manner by the oxidative stress inducer hemin. In addition, biphasic expression with transient upregulation of miR-214 was detected during erythroid differentiation in HbH and β-thalassemia/HbE cells. This implies that miR-214 is involved in the regulation of oxidative stress in both HbH and β-thalassemia/HbE erythroid precursors. However, the effect of miR-214 may differ in late-stage erythroid cells. Our previous study showed a significant difference in the role of miR-214 in oxidative stress regulation between HbH disease and β-thalassemia/HbE in reticulocytes, which appears in the late stage of erythroid cells [13]. The significantly higher expression of miR-214 in reticulocytes was associated with the higher ROS levels in HbH

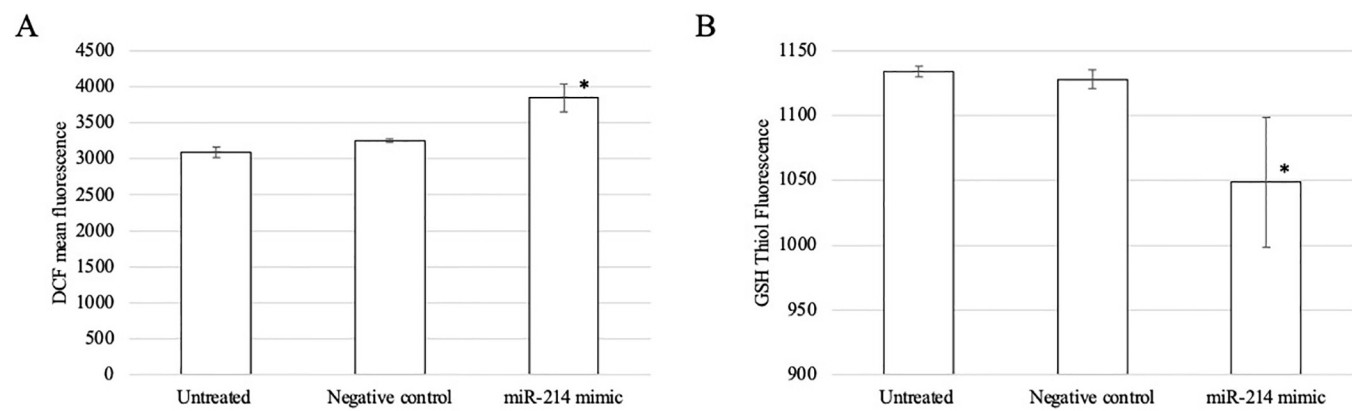

**Fig 7. qPCR analysis of *ATF4* expression after transfection with miR-214 mimic in (A) normal samples and qPCR analysis of the *ATF4* expression after transfection with anti-miR -214 inhibitor in (B) β-thalassemia/HbE and (C) HbH disease.** The experiments were carried out in triplicates. **$p < 0.01$, *$p < 0.05$.

**Fig 8. (A) DCF mean florescence and (B) GSH thiol green florescence of normal erythroblast after transfection with miR-214 mimic.** The data are shown as mean fluorescent intensity. **$p < 0.01$, *$p < 0.05$.

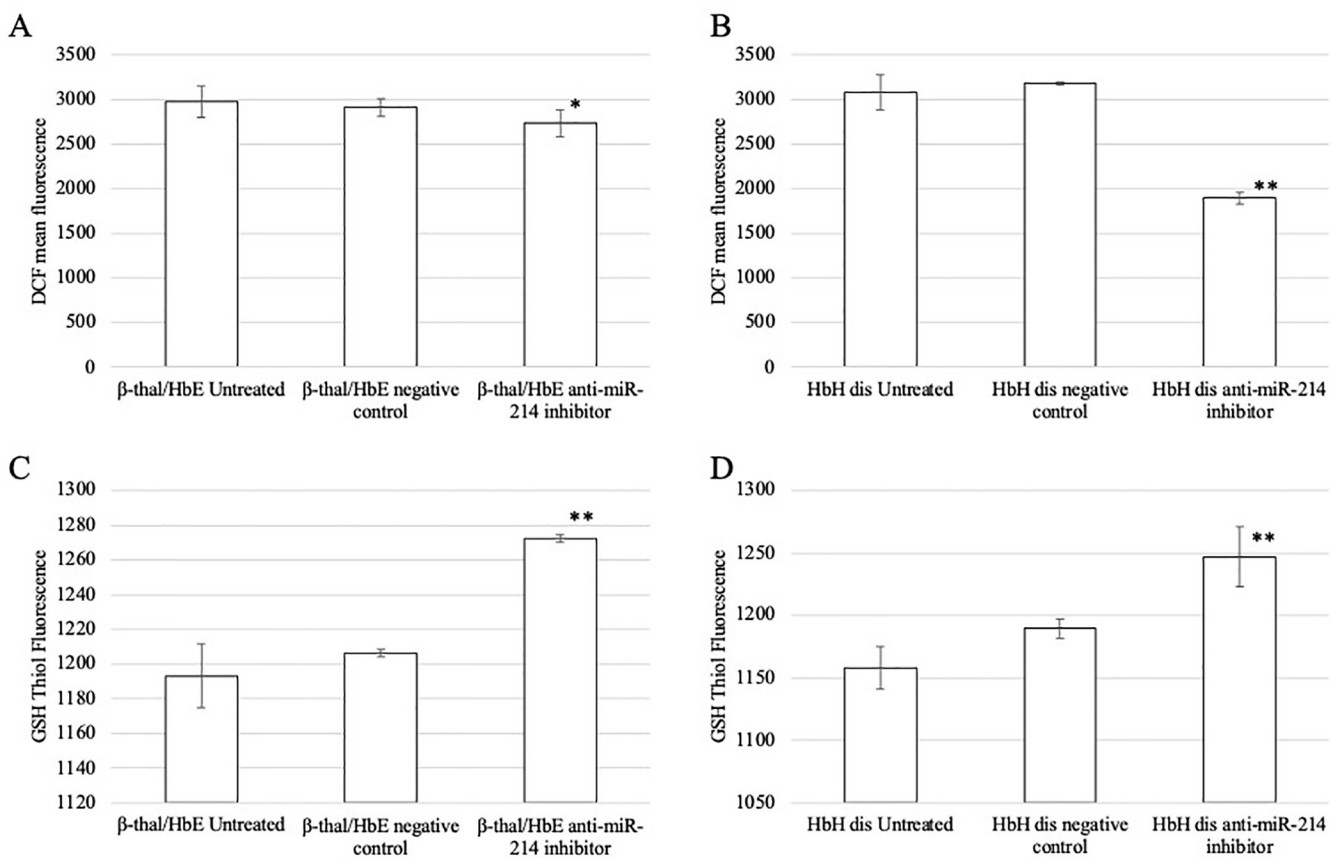

**Fig 9. DCF mean florescence of (A) β-thalassemia/HbE and (B) HbH disease and the GSH thiol green florescence of (C) β-thalassemia/HbE and (D) HbH disease after transfection with anti-miR -214 inhibitor.** The data are shown as mean fluorescent intensity. $^{**}p < 0.01$, $^{*}p < 0.05$.

disease when compared to β-thalassemia/HbE. In the present study, on day 15 of the culture, most of the cultured erythroid cells were polychromatic and orthochromatic erythroblasts. Therefore, no significant differences in miR-214 levels were observed between HbH disease and β-thalassemia/HbE.

ATF4 promotes the transcriptional activation of genes involved in amino acid metabolism, including GSH biosynthesis [15]. High ATF4 expression was observed when a heme-regulated inhibitor of protein translation (HRI) phosphorylated eukaryotic translation initiation factor 2alpha (eIF2alpha) to diminish oxidative stress in erythrocytes [14, 16]. *ATF4* is a mRNA target of miR-214 [13, 25]. In our study, downregulation of *ATF4* mRNA level was observed in both the thalassemia groups—this implied that miR-214 might lead to downregulation of ATF4, which could further promote oxidative stress.

In conclusion, our data demonstrated that under oxidative conditions, the levels of miR-214 increased. However, oxidative stress can also activate ATF4, a putative target of miR-214. Studies on oxidative stress regulation by miR-214 were performed by transfecting miR-214 into cells with hemin-induced oxidative stress and we found that miR-214 could inhibit ATF4 expression under oxidative stress conditions. Subsequently, we examined miR-214 expression in thalassemia erythroid progenitor cell cultures. Biphasic expression with transient upregulation of miR-214 was observed on day 5 of erythroid differentiation in HbH disease and β-thalassemia/HbE. Our study revealed that the loss of function of miR-214 leads to a reduction in

translational activation of *ATF4* mRNA and a subsequent decrease in ROS levels and increase in GSH levels in thalassemia erythroid cells. Taken together, our findings suggest that a significant molecular mechanism to reduce oxidative stress may involve the miR-214-*ATF4* regulatory pathway in erythrocytes with thalassemia. In the future it would be interesting to determine whether miR-214 dysregulation plays a crucial function in oxidative stress status and its involvement in hemolytic susceptibility in other hemolytic diseases.

## Supporting information

**S1 Fig. Proliferation and differentiation analysis of the cultured erythroid progenitors from normal subjects and patients with HbH and β-thalassemia/HbE disease.** (A) Cell number of the cultured erythroid progenitor cells (B) Analysis of erythroid differentiation by flow cytometry (C) Morphology of erythroid differentiation by Wright-Giemsa staining. (TIFF)

**S1 Table. Quantitative PCR analysis data of miR-214 expression normalized against RNU48 expression ($2^{-\Delta Ct}$) after transfection with miR-214 mimic in normal samples and after transfection with anti-miR -214 inhibitor in β-thalassemia/HbE and HbH disease.** (DOCX)

## Author Contributions

**Conceptualization:** Kanitta Srinoun.

**Data curation:** Tipparat Penglong, Kanitta Srinoun.

**Formal analysis:** Tipparat Penglong, Kanitta Srinoun.

**Funding acquisition:** Kanitta Srinoun.

**Investigation:** Apisara Saensuwanna, Natee Jearawiriyapaisarn, Kanitta Srinoun.

**Methodology:** Apisara Saensuwanna, Husanai Jantapaso, Pongpon Phuwakanjana, Natee Jearawiriyapaisarn, Kittiphong Paiboonsukwong, Worrawit Wanichsuwan, Kanitta Srinoun.

**Resources:** Kittiphong Paiboonsukwong.

**Software:** Pongpon Phuwakanjana.

**Writing – original draft:** Kanitta Srinoun.

**Writing – review & editing:** Tipparat Penglong, Apisara Saensuwanna, Husanai Jantapaso, Pongpon Phuwakanjana, Natee Jearawiriyapaisarn, Kittiphong Paiboonsukwong, Worrawit Wanichsuwan.

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
