## [Decision Letter · Decision Letter 0]

11 Jan 2024

PONE-D-23-27864miR-214 aggravates oxidative stress in thalassemic erythroid cells by targeting ATF4PLOS ONE

Dear Dr. Srinoun,

Thank you for submitting your manuscript to PLOS ONE. After careful consideration, we feel that it has merit but does not fully meet PLOS ONE’s publication criteria as it currently stands. Therefore, we invite you to submit a revised version of the manuscript that addresses the points raised during the review process.

We look forward to receiving your revised manuscript.

Kind regards,

Hossain Uddin Shekhar, Ph.D.

Academic Editor

PLOS ONE

“This work was supported by the National Science, Research and Innovation Fund (NSRF), Prince of Songkla University (Grant No. MET6505130S), the Faculty of Medical Technology, Prince of Songkla University, and Mahidol University (Basic Research Fund: fiscal year 2021)”

“This work was supported by the National Science, Research and Innovation Fund (NSRF), Prince of Songkla University (Grant No. MET6505130S), the Faculty of Medical Technology, Prince of Songkla University, and Mahidol University (Basic Research Fund: fiscal year 2021).”

“This work was supported by the National Science, Research and Innovation Fund (NSRF), Prince of Songkla University (Grant No. MET6505130S), the Faculty of Medical Technology, Prince of Songkla University, and Mahidol University (Basic Research Fund: fiscal year 2021)”

Additional Editor Comments:

The topic and findings of this article are quite interesting. There are some minor corrections suggested by the reviewer. I would like to suggest the authors make the necessary corrections suggested by the reviewer and submit the revised manuscript.

Reviewers' comments:

Reviewer's Responses to Questions

**Comments to the Author**

1. Is the manuscript technically sound, and do the data support the conclusions?

Reviewer #1: Yes

2. Has the statistical analysis been performed appropriately and rigorously? 

Reviewer #1: Yes

3. Have the authors made all data underlying the findings in their manuscript fully available?

Reviewer #1: Yes

4. Is the manuscript presented in an intelligible fashion and written in standard English?

Reviewer #1: Yes

5. Review Comments to the Author

Reviewer #1: Authors performed experiments in erythroid progenitors isolated from thalassemia Hb H and beta/Hb E patients, and normal subjects, and investigated effect of micro-RNA-214 on the erythroid by monitoring levels of expressions of mi-RNA214 and ATF4, GSH and ROS content, erythroid cell viability and apoptosis. The results are very interesting and clearly understood. I appreciate all the results and discussion; however, I would like to address my comments on the article.

Title: Yes, it's named appropriately.

Abstract and Introduction: There are some redundance that need to be modified.

Methods: I suggest the authors introduce details of the excitation and emission wavelengths used in flow cytometry of GSH and ROS, important!!

References: Use "Sci World J" instead of "Scientific World Journal".

Figure illustration: x-Axis, y-axis and graph lines in figures 6-9 are not clear, please improve their quality.

Good luck.

Best regards

6. PLOS authors have the option to publish the peer review history of their article (what does this mean?). If published, this will include your full peer review and any attached files.

Reviewer #1: No

---

## [Author Response · Author response to Decision Letter 0]

16 Jan 2024

Reviewer #1: Authors performed experiments in erythroid progenitors isolated from thalassemia Hb H and beta/Hb E patients, and normal subjects, and investigated effect of micro-RNA-214 on the erythroid by monitoring levels of expressions of mi-RNA214 and ATF4, GSH and ROS content, erythroid cell viability and apoptosis. The results are very interesting and clearly understood. I appreciate all the results and discussion; however, I would like to address my comments on the article.

Title: Yes, it's named appropriately.

Abstract and Introduction: There are some redundance that need to be modified.

Reply

Thank you for the comments and suggestion. Some redundance was modified.

Methods: I suggest the authors introduce details of the excitation and emission wavelengths used in flow cytometry of GSH and ROS, important!!

Reply We had added the details of the excitation and emission wavelengths. This had been indicated in Materials and methods part, page 9, line 180-182 and line 188-189, 201 and page 10, line 203-204.

References: Use "Sci World J" instead of "Scientific World Journal".

Reply Thank you for the comments and suggestion. The reference had been corrected. 

Figure illustration: x-Axis, y-axis and graph lines in figures 6-9 are not clear, please improve their quality.

Reply Thank you for the comments and suggestion. We had improved the picture quality.

---

## [Decision Letter · Decision Letter 1]

7 Mar 2024

miR-214 aggravates oxidative stress in thalassemic erythroid cells by targeting ATF4

PONE-D-23-27864R1

Dear Dr. Srinoun,

We’re pleased to inform you that your manuscript has been judged scientifically suitable for publication and will be formally accepted for publication once it meets all outstanding technical requirements.

Kind regards,

Gary S. Stein

Academic Editor

PLOS ONE

Additional Editor Comments (optional):

Reviewers' comments:

Reviewer's Responses to Questions

**Comments to the Author**

1. If the authors have adequately addressed your comments raised in a previous round of review and you feel that this manuscript is now acceptable for publication, you may indicate that here to bypass the “Comments to the Author” section, enter your conflict of interest statement in the “Confidential to Editor” section, and submit your "Accept" recommendation.

Reviewer #1: All comments have been addressed

2. Is the manuscript technically sound, and do the data support the conclusions?

Reviewer #1: Yes

3. Has the statistical analysis been performed appropriately and rigorously? 

Reviewer #1: Yes

4. Have the authors made all data underlying the findings in their manuscript fully available?

Reviewer #1: Yes

5. Is the manuscript presented in an intelligible fashion and written in standard English?

Reviewer #1: Yes

6. Review Comments to the Author

Reviewer #1: The authors have addressed and responded to all my comments. I have no doubt and endorse to publish the revised manuscript.

7. PLOS authors have the option to publish the peer review history of their article (what does this mean?). If published, this will include your full peer review and any attached files.

Reviewer #1: No

---

## [Editor Report · Acceptance letter]

3 Apr 2024

PONE-D-23-27864R1 

PLOS ONE

Dear Dr. Srinoun, 

I'm pleased to inform you that your manuscript has been deemed suitable for publication in PLOS ONE. Congratulations! Your manuscript is now being handed over to our production team.

Kind regards, 

on behalf of

Dr. Gary S. Stein 

Academic Editor

PLOS ONE